# Applications of Xerophytophysiology and Signal Transduction in Plant Production—Flower Qualities in *Eustoma grandiflorum* Were Improved by Sub-Irrigation

**Hui-Lian Xu [1,\*], Jianfang Bai [1,2,\*], Saneyuki Kawabata [3] and Tingting Chang [4]**

[1]  School of Biological Science and Technology, The University of Jinan, Jinan 250024, China
[2]  Beijing Academy of Agriculture and Forestry Sciences, Beijing 100097, China
[3]  Graduate School of Agricultural and Life Sciences, The University of Tokyo, Tokyo 113-8657, Japan
[4]  College of Agricultural Science and Engineering, Hohai University, Nanjing 211100, China
\*  Correspondence: huilianxu@mail.com (H.-L.X.); baijianfang131@163.com (J.B.)

**Abstract:** Relatively mild xerophytic or hardening treatments can induce healthy development of plants. In the present study, as one of xerophytophysiological applications, sub-irrigation was applied to a flower plant of *Eustoma grandiflorum* to confirm whether the sub-irrigation improved flowering quality in addition to plant growth and physiology. As shown by the results, long-term sub-irrigation induced osmotic adjustment, with osmolyte concentration increasing 32.8 osmol m$^{-3}$ ($p \leq 0.01$), improved leaf photosynthetic activities, with more than 10% ($p \leq 0.05$) increase in photosynthetic capacity, and promoted plant growth, with a shoot biomass increase by 27.5% ($p \leq 0.01$) and a root increase by 50.5% ($p \leq 0.01$). These improvements were attributed to turgor maintenance and cell water re-compartmentation into the symplasm, which were both the consequence of osmotic adjustment. The lower osmotic potential and lower relative leaf water potential at incipient plasmolysis suggested that plants in sub-irrigation plots might be more resistant to environmental stresses. Sub-irrigation also improved flower quality shown by increased anthocyanin concentration (16% up, $p \leq 0.01$). Flower quality improvement might be attributed to up-regulation of the *PAL* gene, which could catalyze the synthesis of anthocyanins. *PAL* gene up-regulation might be associated with a concentration increase in salicylic acid (SA), which was suggested as a plant hormone for signaling. Sub-irrigation also affected the flower opening and closing oscillations with less changed opening size or oscillation amplitude. We adopted mathematical models and thoroughly analyzed dynamic changes in photosynthesis, plant growth, and flower opening oscillations. In conclusion, sub-irrigation was a feasible practice and could be used in *E. grandiflorum* culture to improve plant growth and flower opening quality.

**Keywords:** circadian clock; *Eustoma grandiflorum*; flower opening; mathematical model; oscillation; osmotic adjustment; PAL (phenylalanine ammonia-lyase); salicylic acid; turgor potential; xerophytophysiology

## 1. Introduction

The new compound word "Xerophytophysiology" refers to to the mechanical, physical, and biochemical functions of plants in adaptation to drought conditions, including soil water deficit, low humidity, salinity, and strong irradiations [1]. For the application of xerophytophysiology in plants, the "drought" is not necessarily the true water stress that would damage the plant, and instead it is just a stimulus signal to the plant to induce internal adjustments that would benefit the plant. In many cases, the treatment just sends a false signal because no real drought is present. As the signal is sent to the molecular regulation system, the related processes inside the plant change in response to the signal [1]. As a change, osmotic regulation makes the concentration of solute in cells higher than usual. The plant is characterized morphologically by more developed roots and thicker leaves, with more wax and cuticle deposited on the surface, and deeper color in leaves and flowers;

it is physiologically characterized by higher leaf turgor potential and higher photosynthetic activities [2,3]. Such a hardened plant will be healthier and have a higher resistance to biological stress and abiotic stress. Many biological processes in plants change in adaptation to imposed "xerophytic stimulations" at the early stages, and the hardened abilities can last throughout the whole life cycle. In the broad sense, the xerophytic factors used to stimulate plants to induce beneficial changes include mild or false soil water deficit, mild low humidity, hypertonia or salinity, and high irradiations, especially by UV and blue light. "Xerophytophysiology" was proposed by Xu and has been applied in many plants. The topic deals with stimulating treatments to plants under well-watered conditions instead of real drought conditions [1]. The study cases of xerophytophysiological applications include the exposure of sorghum mesocotyls [4] and peanut hypocotyls [5], partial root zone drying for crops of tomato [6] and potato [7], restricted irrigation for tomato [8,9], under-canopy blue light irradiation in tomato crop [10], and drying the cut trace of potato seed tubers [11].

Sub-irrigation is one of the restricted irrigation regimes and is widely used to reduce the pollution of field or nursery runoff to soil, surface, and groundwater [12]. Sub-irrigation differs from normal irrigation; it supplies water to plants by capillaries, and there is no water in the soil air space. Instead, water first fills the air space in the soil. Together with lower canopy humidity, the long-term air-saturated soil conditions give a false drought signal to the plants, inducing internal adjustment, such as maintenance of a high leaf turgor potential, high symplast water content, and healthy morphology [9,13]. Thus, sub-irrigation is taken as one of the relatively mild xerophytic stimulations to induce xerophytophysiological regulations [9]. There are many other kinds of practices based on the theory, such as tomato [6] and potato [7] crops, ridged bed with capillary water supply for wheat [14], and drip infiltration irrigation for greenhouse tomato [8]. However, there are few reports showing whether sub-irrigation improves flower crops. Moreover, flower plants such as *Epimedium grandiflorum* always show circadian oscillations in opening and closing rhythm. It is reported that the diurnal opening and closing of *E. grandiflorum* is probably regulated by diurnal changes in light intensity, high or low temperatures, low air humidity, and soil water deficit as well as the endogenous circadian clocks [15,16]. The flower opening and closing rhythm also reflects the flowering quality. High flowering quality is characterized by fully opening for long time and a rich and deep color. Moreover, flowers of *E. grandiflorum* open fully in during the day and close completely at night, showing circadian oscillations. The flowering quality also includes the precise consistency of the flower opening and closing to the circadian clock rhythm. In this study, we investigate osmotic adjustment and flower opening and closing rhythms in *E. grandiflorum* to evaluate if it works on signal transduction and xerophytophysiology. In addition, a mathematical analysis was adopted to elucidate the details of the flower opening and closing oscillations.

Anthocyanins contribute to the coloration in various tissues of plants, including flowers, fruit, and even in leaves. The anthocyanin-related color change is attributed to increases in anthocyanins as a typical xerophytophysiological adaptation. Generally, anthocyanin-rich plants tissues show high resistance to drought stress [17–19]. Moreover, the synthesis of anthocyanins is catalyzed by phenylalanine ammonia-lyase (PAL), which is determined and regulated by its template gene (*PAL*). PAL is the first reaction enzyme that catalyzes biosynthesis of phenylpropanoid. It plays an important role in plant pigment formation, cell differentiation, and resistances to disease, insects, and environmental stresses [20,21]. As the catalytic enzyme for the first step of anthocyanin biosynthesis, PAL could catalyze deamination of L-phenylalanine to produce trans-cinnamic acid and ammonia [22]. In order to confirm whether sub-irrigation induces anthocyanin synthesis that is regulated by the *PAL* gene, in the present study, not only was the anthocyanin concentration measured, but also the activity of PAL and the expression of the *PAL* gene were analyzed.

In addition, salicylic acid (SA) is not only implicated in anthocyanin synthesis, but also involved in plant defense against pathogen attack. The increase in SA concentration is a signal required for inducing anthocyanin synthesis and the systemic acquired

resistance against the pathogens and abiotic stresses. SA is involved in plant endogenous signaling [23] and regulations of plant biological processes such as growth, development, ripening, and flowering [24]. Research has revealed that treatment with SA under drought stress up-regulates 37 proteins [25]. In the present study, SA, as the signaling hormone, was measured to confirm its relation with anthocyanin synthesis. In summary, the main objective of the present study was to clarify the mechanisms of effects of sub-irrigation on growth, flower quality, and flowering quality of *Eustoma grandiflorum,* in aspects of physiological, biochemical, and molecular biological responses.

## 2. Material and Methods

### 2.1. Preparation of Plant Materials

*Eustoma grandiflorum* cv. Azuma-no-murasaki (Sakata Seed Co., Yokohama, Japan) were used in this study. The methods of planting and management follow Bai et al. (2015) [16]. During the flowering stage, two irrigation regimes were designed: (1) CK—over soil surface irrigation and (2) SI—sub-irrigation by filling the dish with water.

### 2.2. Analysis of Leaf Photosynthesis

Six weeks after treatments were started, leaf photosynthesis of *E. grandiflorum* was measured using the LI-6400 system (Licor Inc., Lincoln, NE, USA). In general, leaf photosynthetic rate ($P_N$) increases in an exponential manner in response to *PPF* changes from low to high. The data of $P_N$ at different *PPF* were collected and analyzed using a mathematical equation of the light response curve [26]. The exponential function equation is

$$P_N = P_C\,(1 - e^{-Ki}) - R_D \tag{1}$$

Detailed definitions of the symbols are in Table 1.

**Table 1.** Abbreviations or symbols and their units.

| Abbr. | Unit | Definition |
|:---:|:---:|:---:|
| | | Photosynthesis–light response curve analysis |
| $P_C$ | μmol m$^{-2}$ s$^{-1}$ | The photosynthetic capacity. |
| $P_N$ | μmol m$^{-2}$ s$^{-1}$ | The net photosynthetic rate. |
| $R_D$ | μmol m$^{-2}$ s$^{-1}$ | Dark respiration rate. |
| $K$ | μmol$^{-1}$ m$^2$ s | Constant proportional to the initial slope and the curviness of the photosynthesis–light response curve. |
| $i$ | μmol m$^{-2}$ s$^{-1}$ | Photosynthetic photon flux. |
| $Y_Q$ | mol mol$^{-1}$ | The maximum quantum yield, proportional to the initial slope of the photosynthesis–light response curve, calculated as $Y_Q = KP_C$. |
| | | Pressure–Volume curve analysis |
| $\Psi$ | MPa | Leaf water potential. |
| $\Psi_{FT}$ | MPa | $\Psi$ at fully turgid status. |
| $\pi$ | MPa | Leaf osmotic potential. |
| $\pi_{s+a}$ | MPa | The average osmotic potential ($\pi$) with hypothesis that symplastic solution was diluted by apoplast water. |
| $\pi_{FT}$ | MPa | $\pi$ at fully turgid status. |
| $\pi_{IP}$ | MPa | $\pi$ at zero turgor or incipient plasmolysis, calculated from the P-V curve at $\Psi = 0.99\pi$. |
| $\zeta_{FT}$ | | Relative leaf water content ($\zeta$) at fully turgid status. |
| $\zeta_{IP}$ | | $\zeta$ at the zero-turgor point. |
| $\zeta_{ap}$ | | The water fraction in apoplasm. |

**Table 1.** *Cont.*

| Abbr. | Unit | Definition |
|---|---|---|
| $\zeta_{sym}$ | | The water fraction in symplasm. |
| $\alpha$ | | Constant proportional to the slope of the initial part of the P-V curve. |
| $\beta$ | | Constant proportional to the slope of the second sloping phase of the P-V curve. |
| $C_{FT}$ | $mol\,m^{-3}$ | Concentration of osmotically active substances. |
| $c\Delta C_{FT}$ | | The active increment of $C_{FT}$ compared with control. |
| | | *Analysis of the transpiration declining curve in excised leaves* |
| $\zeta_{sat}$ | | Relative leaf water content ($\zeta$) at saturation status. |
| $\zeta_{SC}$ | | $\zeta$ at the time when stomata are completely closed. |
| $t$ | | Time since the drying process started. |
| $\alpha'$ | | Constant proportional to slope of the initial steep-sloped part of the curve and to the rapid rate of water loss, mainly by stomatal transpiration. |
| $\beta'$ | | Constant proportional to the slope of the second gently sloped part and to the rate of water loss by cuticular transpiration when stomata are closed. |
| $\tau_D$ | | Time used to dry up the excised leaf to its relative water content of 10%. |
| | | *Analysis of flower opening and closing oscillations* |
| ZT | h | Zone time; in the present study the zone time is Tokyo time. |
| $Y$ | | The opening extent relative to the maximum opening of the flower. |
| $Y_A$ | | The initial amplitude of oscillations. |
| $Y_R$ | | The residual values of $Y$ at the initial oscillation bottom. |
| $Y_{A96}$ | | The amplitude of oscillations at ZT96. |
| $T$ | h | The oscillation period. |
| $\omega$ | | The angular velocity in the sinusoidal function equation. |
| $\lambda$ | | The coefficient to adjust the change of $\omega$. The value of $\omega$ would be larger and thus the oscillation period ($T$) smaller if $\lambda$ was positive; if $\lambda$ was negative, $\omega$ would become smaller and thus $T$ larger as time progressed. |
| $\tau$ | h | The time needed to adjust $Y$ to move into the oscillation process. |
| $\alpha$ | | The coefficient related to the expansion or decay of the oscillation amplitude ($Y_A$), which would be smaller and smaller if $\lambda$ was negative and larger and larger if $\lambda$ was positive. |
| $\beta$ | | The coefficient related to the dynamic change of $Y_R$, which would be larger and larger if $\lambda$ was positive, showing an upward drifting pattern of oscillation rhythm. If $\lambda$ was negative, the oscillation rhythm shows a downward drifting pattern. |
| $f$ | $h^{-1}$ | Oscillation frequency ($f = \omega/2\pi = 1/T$). |
| $T$ | h | The period of oscillation ($T = 2\pi/\omega$). $T_0$ and $T_{96}$ are T at the initial and at ZT96. |

### 2.3. Measurement of Leaf Color

Here, leaf color was used as an indicator of chlorophyll concentration. The leaf color was measured by the SPAD-502 chlorophyll meter (Minolta Camera Co., Osaka, Japan). Many studies have suggested that the SPAD value is related to the chlorophyll concentration of leaves, and the change trend of chlorophyll concentration is positively correlated to the SPAD value of plant leaves.

### 2.4. Dry Mass Production

The shoot and root were separately harvested, dried at 80 °C for 48 h, and the dry mass was recorded. Then, the root/total plant ratio was calculated.

## 2.5. Measurement of Concentration of Anthocyanins

A sample of 0.2 g was taken from the fully expanded petal then treated following the method described by Yang et al. (2022). Then, the relative concentration of anthocyanins was estimated at 530 nm using a spectrophotometer (U-2000, Hitachi) [27].

## 2.6. Determination of Salicylic Acid Concentration

When the flowers were in full bloom, petals were sampled, with each sample of 0.2 g fresh weight. The free salicylic was measureed according to the method of Deng et al. [28]. The sample of 0.2 g fresh petal was ground with 3 mL 90% methanol in an ice bath and centrifuged at $12,000 \times g$ for 15 min. The precipitation was dissolved with 3 mL of 100% methanol, centrifuged again, and mixed with the previous supernatant. The mixed supernatant of 5 mL in each tube was centrifuged at $1500 \times g$ for 10 min, and the supernatant was dried at 40 °C by rotation. Then, 3 mL of $ddH_2O$ were added in and dissolved at 80 °C for 10 min. A sample of 1 mL was extracted by adding 25 mL ethyl acetate-cyclohexane (1:1) and 50 μL concentrated hydrochloric acid. The upper liquid was collected and dried with nitrogen gas and dissolved with 1 mL 20% methanol (20 mM sodium acetate, at pH 5) for the determination of free salicylic acid. The HPLC conditions were as follows: 20% methanol solution (20 mM sodium acetate buffer, at pH 5) was used as mobile phase; the flow rate was 1 mL min$^{-1}$, the liquid chromatography column was ODS120 ($4.6 \times 250$ nm), the detector was a fluorescence detector, the excitation wavelength was 295 nm, and the emission wavelength was 370 nm.

## 2.7. PCR Analysis for PAL Gene Expression

Analysis of the expression of the gene *PAL* was performed by quantitative PCR. The PCR primer of the *PAL* gene was F-5′-AAGCC GAAACAAGATAGATAC-3′/R-5′-GTTTACAGAGTTGATTTCCCT-3′. The PCR extracted RNA from petals for reverse transcription to synthesize cDNA. PCR conditions were 95 °C/5 min, 95 °C/10 s, and 60 °C/30 s, for a total of 40 cycles. The PCR reaction system is SYBR® Premix Ex Taq™ II (TaKaRa) 10 μL with an upstream primer of 0.8 μL (0.4 μM) and a downstream primer of 0.8 μL (0.4 μM). The Rox dye is 0.4 μL, the template (cDNA) is 1.0 μL (<50 ng), the $ddH_2O$ is 7 μL, and the total volume is 20.0 μL. Quantitative results were expressed as a ratio to Actin 2 (intrinsic regulatory gene) [29].

## 2.8. Data Collection for Pressure–Volume Curve Analysis

When studying the water status of plants, we used the pressure chamber to measure the equilibrium pressure and the corresponding change in water content of the plant sample (such as a plant leaf or a shoot), and the graph is drawn according to the corresponding relationship between the two variables, the equilibrium pressure imposed on the plant sample and the water volume in that plant at that equilibrium pressure. Because it is difficult to measure the absolute value of water volume, the relative water potential is usually used instead. In the present study, we analyzed osmotic adjustment, turgor maintenance, and cell water compartmenting by the pressure–volume curve technique [30] modified by Xu et al. [31]. A leaf blade of *E. grandiflorum* was excised under water in a container and rehydrated by immersing the cut trace in water for 4 h, when the leaf would be saturated and fully turgid. The relative leaf water content at each pressure increment was measured using a previous method [31], and then it was used as the independent variable (X-axis); the reciprocal of the equilibrated pressure at each increment was used as the dependent variable (Y-axis). The graph drawn using these collected data was called the pressure–volume curve or the P-V curve.

## 2.9. Modeling Equation Used in this Paper

(1)  *An improved modeling equation in the analysis of the pressure–volume curve by* [7]

$$-\Psi^{-1} = \{\Psi_{FT}{}^{-1} - \pi_{s+a}{}^{-1}\ [\zeta_o - \beta\ (1 - \zeta) - \zeta_{ap}]\}e^{-\alpha(1-\zeta)} + \pi_{s+a}{}^{-1}\ [\zeta_o - \beta\ (1 - \zeta) - \zeta_{ap}] \tag{2}$$

(2)　　*The incipient plasmolysis or zero-turgor point* [9]

$$-\pi^{-1} = \pi_{s+a}{}^{-1}\ [\zeta_o - \beta\ (1 - \zeta) - \zeta_{ap}] \tag{3}$$

The incipient plasmolysis point is defined as the point (y, x), i.e., $[(-\Psi^{-1})_{IP}, (1 - \zeta)_{IP}]$, where

$$y = (-\pi^{-1} + \delta) = (-\Psi^{-1} - \delta)\ (\delta \le 0.01y) \tag{4}$$

The zero-turgor point at a lower osmotic potential or at a lower relative leaf water content means a higher water stress tolerance.

(3)　　*The re-compartmentation of the symplastic and apoplastic water*

The point where the aforementioned linear line in the P-V curve crosses the abscissa (X-axis) is the separatrix between symplastic and apoplastic water fractions. The proportion of water from symplasm and apoplasm will be changed due to the change of concentration of osmolytes inside the cell membrane. Usually, water in symplasm, where many metabolisms exist, is more important than water in apoplasm [32,33].

(4)　　*Analysis of osmotic adjustment*

The equation of osmotic potential ($\pi$) is shown as follows [9]

$$\pi = -RTC_S \text{ or } C_S = -\pi/RT \tag{5}$$

where $C_S$ is solute concentration and $RT$ is 2437 J mol$^{-1}$. The detailed derivation of this algorithm was presented by Xu et al. (2011) [9].

(5)　　*The analysis of leaf water retention ability* [34]

$$\zeta = [\zeta_0 - \zeta_{SC}\ (1 - \beta't)]\ e^{-\alpha't} + \zeta_{SC}\ (1 - \beta't) \tag{6}$$

Water retention ability (WRA) is expressed as

$$WRA = 1/\alpha + 1/\beta \tag{7}$$

(6)　　*Analysis of the diurnal opening and closing oscillations*

Flowers began to open four weeks after the seedlings were treated with sub-irrigation. The diameter of the same flower was measured eight times a day at 5:00, 8:00, 10:00, 12:00, 15:00, 18:00, 20:00, and 24:00. The rhythmic phenomenon of flower opening and closing was reflected by the relative diameter of the flower. The relative diameter ($D_R$) was calculated as

$$D_R = (D_t - D_{min})/(D_{max} - D_{min}) \tag{8}$$

As shown in Figure 1, $D_t$ was the value of the flower diameter ($D$) at a measuring time, $D_{max}$ was the maximum value of $D$ at the full opening status, and $D_{min}$ was the minimum value of $D$ at the closed status. An increase in $D$ means flower opening. When $D$ shows its maximum or minimum value, the flower is fully opened or fully closed. The methodology is recorded in detail by Bai in her PhD dissertation [35].

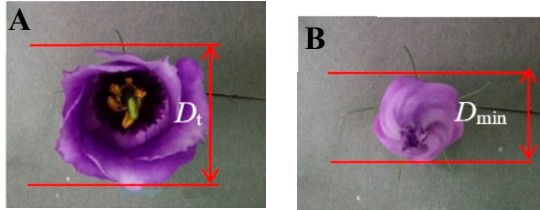

**Figure 1.** Flower pictures to show the opening (**A**), closed (**B**), and full opened status (**C**).

(7)  *Mathematical modeling for the flower opening oscillation curve*

Opening and closure of many flowers show rhythmic oscillations in accordance with the circadian time. The ideal and theoretical pattern of the oscillation process would be shown as in Figure 2 and mathematically modeled by a trigonometric function as modified by Xu et al. [36]:

$$Y = 0.5\text{COS}(\omega t) + 0.5 = 0.5\text{COS}((2\pi/T)t) + 0.5 \tag{9}$$

where $Y$ is the relative position of the petal distal end and $T$ (24 h) is the standard circadian hours of a day.

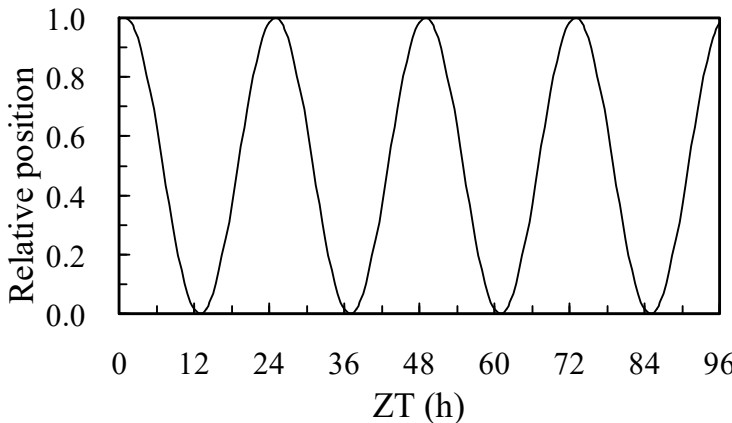

**Figure 2.** An ideal and theoretical pattern of the flower opening and closing oscillation cycles.

The actual pattern of the flowering oscillations changed according to lighting and other conditions as time progressed and could be modeled by a modified trigonometric function as

$$Y = [Y_A + Y_R + Y_A\cos(\omega(1 + \lambda(t + \tau))(t + \tau))](1 + \alpha(t + \tau)) + Y_R(1 + \beta(t + \tau)) \tag{10}$$

The term $(1 + \lambda(t + \tau))$ is used to adjust $\omega$ because it usually changes with time. The term $\alpha(t + \tau)$ is used to adjust the oscillation shown by $[Y_A + Y_R + Y_A\cos(\omega(1 + \lambda(t + \tau)))]$. If the oscillation cycle is diminished gradually as time passes, the value of $\alpha$ is negative, and if amplified gradually, the value of $\alpha$ is positive. In the same way, the term $(1 + \beta(t + \tau)$ is used to adjust the bottom value ($Y_R$).

In the equation, $\omega$ is the angular velocity of the oscillation and defined as

$$\omega = 2\pi f = 2\pi/T \tag{11}$$

where $f$ is the oscillation frequency and $T$ is the oscillation period.

The values of $\omega$ and $T$ at ZT 96 h were calculated as

$$\omega_{96} = \omega_0(1 + \lambda t) \ (t = 96) \tag{12}$$

and

$$T_{96} = 2\pi/\omega_{96} \ (t = 96) \tag{13}$$

Similarly, $Y_A$ at ZT96 was calculated as

$$Y_{A96} = Y_A (1 + \alpha t) \ (t = 96) \tag{14}$$

and

$$Y_{R96} = Y_R (1 + \beta t) \ (t = 96) \tag{15}$$

The symbols and definitions are presented in Table 1. More details can be found in a similar mathematical analysis used for peanut plant stomatal oscillations [37].

Abbreviations or symbols and their units are showed in Table 1.

### 2.10. Statistical Analysis

Measurement was made for five flowers in each lighting regime. The typical three close to average were plotted together with the mathematical modeling curve in the figures. The variables of the modeling curve were collected from each measurement, and the average of the variables was presented in the tables. The data were analyzed by ANOVA using SPSS software (V22.0). Significance was considered as $p \leq 0.05$ and $p \leq 0.05$, and no significance was marked as *, **, and ns, respectively, in Tables 2–5.

**Table 2.** Photosynthetic capacity ($P_C$), dark respiration ($R_D$), the maximum quantum yield ($Y_Q$) and growth and yield factors as well as leaf color (SPAD), the relative anthocyanin (Antho) concentration, salicylic acid concentration (SA), and relative expression of *PAL* gene.

| Irrigation | $P_C$ | $RD$ | $Y_Q$ | Biomass (g pl$^{-1}$) | | | R/T | Leaf Color | Antho | SA | PAL |
|---|---|---|---|---|---|---|---|---|---|---|---|
| | ($\mu$mol m$^{-2}$ s$^{-1}$) | | (mol mol$^{-1}$) | Shoot | Root | Total | (%) | (SPAD) | (A530 g$^{-1}$ FW) | ($\mu$g kg$^{-1}$) | |
| Overhead | 18.6 | 2.7 | 0.0452 | 7.02 | 0.95 | 7.97 | 11.9 | 37.9 | 68.3 | 3.2 | 1.17 |
| Sub | 20.7 | 3.0 | 0.0534 | 8.95 | 1.43 | 10.38 | 13.8 | 41.2 | 79.2 | 44.3 | 7.82 |
| Statistic | * | * | ** | * | ** | ** | * | ** | ** | ** | ** |

Significance at $p \leq 0.05$ and $p \leq 0.01$ is marked as * and **, respectively. See Table 1 for abbreviation symbols and their units.

**Table 3.** Parameters obtained from the P-V curve analysis for plants of *E. grandiflorum* under different irrigation regimes.

| Irrigation | $\Psi_{FT}$ | $\pi_{FT}$ | $P_{FT}$ | $\pi_{s+a}$ | $\Psi_{MD}$ | $\pi_{MD}$ | $P_{MD}$ | $\pi_{IP}$ | $\zeta_{sym}$ | $\zeta_{apo}$ | $\alpha$ | $\beta$ | $\zeta_{IP}$ | $C_{FT}$ | $\Delta C_{FT}$ |
|---|---|---|---|---|---|---|---|---|---|---|---|---|---|---|---|
| Overhead | −0.27 | −0.88 | 0.61 | −0.57 | −0.65 | −0.99 | 0.34 | −1.11 | 0.26 | 0.74 | 47.6 | 0.98 | 0.842 | 360.8 | 0.0 |
| Sub | −0.28 | −0.96 | 0.68 | −0.63 | −0.63 | −1.08 | 0.43 | −1.19 | 0.31 | 0.69 | 54.3 | 0.99 | 0.821 | 393.6 | 32.8 |
| Statistic | ns | ** | ** | ** | ns | ** | ** | * | ** | ** | * | ns | * | ** | ** |

* and ** mean significance at $p \leq 0.05$ and $p \leq 0.01$, respectively; ns means no significance. See Table 1 for the symbols and their units.

**Table 4.** Parameters of the transpiration decline curve in the excised leave of *E. grandiflorum* under different irrigation regimes.

| Irrigation | $\zeta_{SC}$ | $\alpha$ | $\beta$ | $\tau_D$ (10$^3$ s) | WRA |
|---|---|---|---|---|---|
| Overhead | 0.721 | 0.648 | 0.122 | 70.6 | 9.74 |
| Sub | 0.702 | 0.699 | 0.101 | 84.1 | 11.33 |
| Statistic | * | * | ** | ** | ** |

* and ** mean significance at $p \leq 0.05$ and $p \leq 0.01$, respectively; ns means no significance. See Table 1 for the symbols and their units.

**Table 5.** Parameters from the analysis of flower opening and closing oscillations in *E. grandiflorum* under different irrigation regimes in natural light conditions.

| Plot | $\omega$ | $Y_A$ | $Y_{A96}$ | $Y_R$ | $T_0$ | $T_{96}$ | $\lambda$ | $\tau$ | $\alpha$ | $\beta$ |
|------|------|------|------|------|------|------|------|------|------|------|
| Overhead | 0.252 | 0.426 | 0.253 | 0.03 | 24.92 | 24.30 | −0.00454 | 6.2 | −0.000284 | 0.0891 |
| Sub | 0.255 | 0.438 | 0.326 | 0.03 | 24.63 | 24.12 | −0.00284 | 6.1 | −0.000235 | 0.0778 |
| Statistic | * | * | * | ns | * | * | ** | ns | * | * |

* and ** mean significance at $p \leq 0.05$ and $p \leq 0.01$, respectively; ns means no significance. See Table 1 for the symbols and their units.

## 3. Results

### 3.1. Sub-Irrigation Improved Leaf Photosynthetic Activities, Plant Growth, and Flower Quality

#### 3.1.1. Photosynthetic Activities

Photosynthetic capacity ($P_C$) is the maximum of the gross photosynthetic rate, with the respiration rate included, at the saturated light intensity. $P_C$ shows the photosynthetic potential of a plant. In this experiment, $P_C$ was increased 11.3% by the sub-irrigation treatment, and the significance reached the $p \leq 0.05$ level. The maximum quantum yield ($Y_Q$) shows the potential light use efficiency of a plant. In the present experiment, $Y_Q$ was increased 18.1% by sub-irrigation treatment, and the significance reached the $p \leq 0.01$ level. The respiration rate ($R_D$) in the dark can reflect the strength of metabolic activities of a plant. In this experiment, $R_D$ was increased more than 10% by sub-irrigation and reached the $p \leq 0.05$ level.

#### 3.1.2. Biomass Production

Because the photosynthetic activities were improved, the plant biomass was also increased by sub-irrigation. Compared with the control plot of overhead irrigation, in the sub-irrigation plot, shoot biomass increased 27.5% ($p \leq 0.01$), root biomass increased 50.5% ($p \leq 0.01$), and consequently, the root/shoot ratio significantly increased. When a plant perceives its rhizosphere to be under drought, it always tries to enlarge its root system for as much water as possible. Here, the increased root/shoot ratio was an indication of adaptation to drought. In the present study, the treatment of sub-irrigation was not a real drought factor, and it was used to maintain the soil aeration without water filled. It gave plants the illusion of drought.

#### 3.1.3. Leaf Color and Flower Quality

In the sub-irrigation plot, the plants also showed a healthy appearance, with deeper color in leaves and flowers, which was also confirmed by higher concentrations of leaf chlorophyll (3.3 point up) and higher flower anthocyanin (16% up) (Table 2). In particular, higher anthocyanin concentration in the flower meant better flower.

### 3.2. Increase in Salicylic Acid Concentration and Up-Regulation of PAL Gene

#### 3.2.1. Concentration of Salicylic Acid

Salicylic acid (SA) is a sensitive signal hormone in response to environmental stresses. In this study, SA increased to 14 times compared to the control plot. The sub-irrigation adopted in the present experiment was just a mild and false drought, and it could just keep the soil highly aerated with sufficient water supply by capillary connections. The aerated space between soil particles might give the plant a false signal of a water deficit. It might be the false signal that induced a sharp increase in SA concentration, while it might be the high concentration of the signal compound SA that induced up-regulation of the *PAL* gene and the consequent increase in anthocyanin concentration.

#### 3.2.2. Up-Regulation of PAL Gene

The extent of *PAL* gene regulation was expressed as a ratio to Actin 2. As shown in Table 2, the *PAL* gene was up-regulated to 7 times the levels of the control plot. This

was consistent with the concentration of SA but not necessarily consistent with anthocyanin concentration. Nevertheless, as the consequence of *PAL* up-regulation, anthocyanin concentration increased to a significant level ($p \leq 0.01$). It was suggested that the SA concentration increase, PAL up-regulation, and the anthocyanin concentration increase are serially correlated.

*3.3. Sub-Irrigation Induced Osmotic Adjustment and Improved Leaf Turgor Potential*

3.3.1. Leaf Turgor Potential and Osmotic Adjustment

Leaf turgor potential ($P$) is the driving force for plant growth, mainly in the form of cell enlargement. $P$ is also a positive factor related to stomatal opening for photosynthesis. Maintenance of $P$ is especially important in adverse environmental conditions, such as in water deficit and at hot dry midday. $P$ is the difference between leaf water potential ($\Psi$) and the leaf osmotic potential ($\pi$), i.e., $P = \Psi - \pi$. The value of $P$ is always positive, but $\Psi$ and $\pi$ are always negative. Thus, at a given $\Psi$, the lower (more negative) the value of $\pi$, the higher the value of $P$. Here, it must be noticed that, when the term of osmotic pressure instead of potential is used, the value must be shown to be positive. When leaf $\Psi$ is unchanged while $\pi$ is lowered in response to environmental changes, the phenomenon is called osmotic adjustment, i.e., the concentration of solutes in cells is actively increased to resist the adverse environment. The parameters from the P-V curve analysis are shown in Table 3. Osmotic potential at fully turgid status ($\pi_{FT}$), at midday ($\pi_{MD}$), and at the plasmolysis point ($\pi_{IP}$) were all lower in *E. grandiflorum* leaves in the sub-irrigation than in the overhead irrigation treatments. Therefore, the leaf turgor potential at fully turgid status ($P_{FT}$) and at midday ($P_{MD}$) was higher in the sub-irrigation treatment, while the water potential at full turgor ($\Psi_{FT}$) and at midday ($\Psi_{MD}$) were at similar levels. Active increases in osmotic solute concentrations shown by the value of $\Delta C_{FT}$ accounted for decreases in osmotic potential. The aforementioned results suggested that osmotic adjustment really occurred in plants subjected to sub-irrigation.

3.3.2. Cell Water Compartmenting

As the whole cell, water is stored in symplasm (cytoplasm) and apoplasm (mainly cell walls) in a ratio of about 75:25%. In water-stressed conditions, osmotic concentration increases in symplasm, and then water in apoplasm moves to symplasm, where there are more metabolic processes. In the present experiment, at least as one of the reasons was that increased osmotic solute concentration in symplasm caused the inward flow of water from apoplasm to symplasm, which was confirmed by the values of $\zeta_{sym}$ and $\zeta_{apo}$. The total water content of the plant tissue is divided into symplastic and apoplastic fractions by the plasmodesmata. Symplastic water is the water contained within cell membranes. Since symplastic water is the water in the cell membrane, the water potential can be determined by the osmotic and turgor potential. While apoplastic water is the water that exists in intercellular space and the cell wall, the water potential is only affected by the osmotic component. Some studies suggested that the water in the xylem is apoplastic water, since the xylem cells are dead at maturity and have no membranes. Thus, the increase in symplastic water fraction is one of the consequences of the xerophytophysiological regulations. For example, the plant may re-compartment the cell water from 70 to 73.4% in the symplasm and 30 to 21.6% in the apoplasm when exposed to environmental stress. Studies suggested that a cell water content of 73.4% in the symplasm will be beneficial to the improvement of the related biochemical metabolism and cell turgor potential [32].

3.3.3. Osmotic Potential and Relative Water Content at Incipient Plasmolysis

When water is lost to a large enough extent, the cell wall separates from the cell membrane. At this point, $P$ is 0 and the leaf wilts. This phenomenon is called "incipient plasmolysis". At the same level, a plant with high water stress resistance shows lower osmotic potential ($\pi_{IP}$) and lower relative water content ($\zeta_{IP}$). In the present experiment, lower values of $\pi_{IP}$ and $\zeta_{IP}$ might show the desiccation stress tolerance in leaves

of *E. grandiflorum* in the sub-irrigation treatment. In all the parameters, increased turgor potentials and the symplast water fraction (Table 3) might be directly associated with improved photosynthetic activities and plant growth (Table 2).

### 3.4. Leaf Water Retention Ability

Water loss by transpiration would continue if the rehydrated excised leaves were placed under lighting. As the leaves lost water, stomata closed in response to the leaf water deficit. The relative water content of leaves ($\zeta$) decreased sharply with the increase in stomatal transpiration, showing a pattern of minus exponential function. The minus exponential decreasing curve can be modeled as the aforementioned Equation (10): ($\zeta = [\zeta_0 - \zeta_{SC} (1 - \beta' t)]\, e^{-\alpha' t} + \zeta_{SC} (1 - \beta' t)$). In this curve analysis, in addition to $\zeta_{SC}$ (relative leaf water content at the time point when stomata are completely closed) and $\zeta_o$ ($\zeta$ at full turgor), there were two important constants, $\alpha$ and $\beta$, which were, respectively, proportional to stomatal and cuticular transpiration rates. As shown by the value of $\alpha$ in Table 4, stomatal transpiration was higher but the cuticular transpiration was lower in *E. grandiforum* leaves in the sub-irrigation rather than in the overhead irrigation treatments. The higher stomatal transpiration may be due to the higher stomatal conductance of sub-irrigated plants during leaf dehydration. Moreover, as shown by the value of $\beta$, the lower cuticular transpiration may be due to a thickened cuticular layer and/or more wax deposit on the leaf surface. The value of $\zeta_{SC}$ showed the leaf relative water content when stomata were completely closed and was lower in leaves of sub-irrigated plants than in leaves of control plants, which might be responsible for a higher water deficit tolerance. The constant $\tau_D$ showed the time theoretically used to dry leaves to its relative water content of 10%. Longer water retention duration was confirmed by the higher value of $\tau_D$ in leaves of sub-irrigated plants. Shown as WRA = $\alpha^{-1} + \beta^{-1}$, the water retention ability (WRA) was higher in plants subjected to sub-irrigation than plants in control plots.

### 3.5. Flower Opening and Closing Oscillations

*E. grandiflorum* flowers showed opening and closing rhythmic movements in response to natural light and dark phases. The flower opened fast immediately after illumination by natural light in the morning (Figure 2), reaching the maximum at 10 am, maintaining almost full opening for 5 h, and then closed steadily. As shown in Figure 3, points of measured data did not exactly meet the modeled theoretical curves, but the oscillation bottoms, which showed the flower full closing, fitted the models. As shown in Table 5, the amplitude at initial (YA) was higher in the sub-irrigation treatment. As shown by the values of YA96 (opening extent at the time point of 96 h), the decrease in YA as time progressed was less under the sub-irrigation regime. This meant that sub-irrigation could maintain the flower opening and closing as normal compared with the overhead irrigation regime. This was also seen from the values of $\alpha$ and $\beta$, which were both smaller under the sub-irrigation regime. The oscillation period at both the initial ($T_0$) and at the 96th hour ($T_{96}$) was shorter than but reached more closely to the normal 24 h circadian clock. Under both overhead and sub-irrigation regimes, the oscillation period got a little shorter as time progressed, beginning from $T_0$ as 24.63 and 24.92, and reaching $T_{96}$ as 24.12 and 24.30, respectively. The value of $\lambda$ showed the changing extent of the oscillation period and was less negative under the sub-irrigation regime than under the overhead irrigation regime. The value of $\alpha$ showed the diminishing extent of the oscillations as time progressed. The more negative $\alpha$ was, the larger was the diminishing extent (Figure 3). The larger (more positive) the value of $\beta$, the larger the increasing extent of the bottom value (YR). The value of $\alpha$ was less negative and $\beta$ was less positive in sub-irrigation plots than in control plots. This result suggested that sub-irrigation maintained flowering as normal with less senescence. Overall, sub-irrigation significantly changed the characteristics of *E. grandiflorum* stomatal opening and closing oscillations, i.e., the oscillation amplitude increased, and the oscillation period decreased, even if only to a minor extent. This meant that flowers of *E. grandiflorum*

in the sub-irrigation treatment became more sensitive and more active in response to the xerophytophysiological stimulation by sub-irrigation.

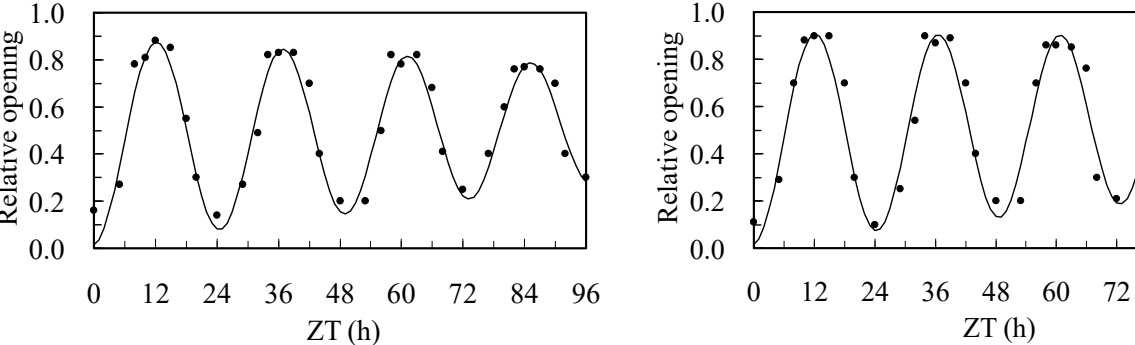

**Figure 3.** Flower opening and closing oscillations in *E. grandiflorum* under overhead (**left**) and sub-irrigation (**right**) regimes in natural light conditions.

## 4. Discussion

Mechanisms for signal transduction in plants subjected to adverse environmental conditions have been studied in detail by using *Arabidopsis* as the model plant [38]. However, these research achievements have not been practically used in plant production. Xu has tried a new research field to use artificial xerophytic stimulations including soil water deficit, strong irradiation (blue light and UV), and low humidity to induce plant health and crop improvement [39]. The stimulations are usually mild, temporary, or partial without real stresses to the plants, aiding in crisis signal transduction and xerophytophysiological regulations. In the present study, as one of applications of signal transduction and xerophytophysiology in plant production, sub-irrigation induced osmotic adjustment shown by improved leaf turgor potential, increased symplast water fraction, and the consequent higher photosynthetic activities and higher biomass production. Osmotic adjustment, induced by sub-irrigation, was characterized by lower osmotic potential ($\pi$) at both midday ($\pi_{MD}$) and fully turgid status ($\pi_{FT}$). Because there is no real water stress imposed on the plants, the leaf water potentials ($\Psi_{FT}$, $\Psi_{MD}$) were not different from those in control plots. Therefore, leaf turgor potentials ($P_{FT}$, $P_{MD}$) were maintained higher ($P = \Psi - \pi$) in sub-irrigation plots than in control plots. Since the lower osmotic potential would pull water from the apoplasm (cell walls) into the symplast (inside the membrane), water in the whole cell was re-compartmented, and the water fraction was more in the symplasm than in the apoplasm. Many studies have confirmed that this kind of water re-compartmenting is good for physiological and biochemical metabolisms in the symplasm [4–11,13,14,26,31,32,34,37]. Moreover, the osmotic potential at incipient plasmolysis ($\Psi_{IP}$) (cell walls just begin to separate from membrane) and leaf relative water content at plasmolysis ($\zeta_{IP}$) were lower in sub-irrigation plots than in control plots. This suggested that plants in sub-irrigation plots could tolerate more severe dehydrations and did not completely lose turgor and wilt until lower leaf water content and osmotic potential were present. These are indications of water stress resistance or tolerance.

The leaf color was a deeper green, shown by higher chlorophyll concentration, and flowers was deeper in color as well, supported by higher anthocyanin concentration. The increased quantum yield might be also attributed to the increased chlorophyll concentration. Morphologically, the plants under the sub-irrigation regime appeared stronger than under the overhead irrigation regime. This was also confirmed by leaf water retention ability analyzed from the declining curve of transpiration [34]. The similar effect of sub-irrigation has been found in tomato crops, where fruit quality is improved in addition to increased fruit yield [9]. However, it is the first time that flower quality is improved in addition to the improved photosynthetic activities and plant growth by sub-irrigation. The improved flower quality was caused in part by increased anthocyanins. Many studies have confirmed that anthocyanin synthesis is catalyzed by the enzyme of

phenylalanine ammonia-lyase [22,27,29]. Up-regulation of the gene encoded for phenylalanine ammonia-lyase (*PAL*) is induced by a concentration increase in the signal hormone, salicylic acid [40,41]. Results of the present study suggested that a concentration increase in salicylic acid as a signal hormone induced up-regulation of the *PAL* gene, and the enzyme of phenylamine ammonia-lyase promoted the anthocyanin synthesis. The positive effect of sub-irrigation was also found on the flowering quality. Flowering quality is characterized by fully opening during the day and completely closing at night, with less senescence and a longer opening period, while flower quality refers to the colorfulness, brightness, and beauty as well as a good size and shape. This opening quality can be shown by circadian oscillations of flower opening and closing [15,35]. As reported previously, as senescence of flower petals develops, the maximum opening extent, or the amplitude of the opening and closing oscillation, gets smaller and smaller, and the oscillations appear less sensitive to the light [35]. In the present study, the amplitude of the opening oscillation in flowers under the sub-irrigation regime was maintained constant and coded well with the 24 h circadian clock compared with those in the overhead irrigation regime. These results were confirmed by the parameters from the mathematical analysis of the opening oscillations. $Y_A$ (amplitude at the first oscillation cycle) and $Y_{A96}$ (amplitude at the 96th hour) were both higher under sub-irrigation. $T_0$ (oscillation period of the first oscillation cycle) and $T_{96}$ (oscillation period at the 96th hour) were closer to the 24 h circadian clock. In the present study, we quantified the properties of flower opening and closing oscillations using Xu's modified oscillation equation, originally used to analyze the stomatal opening oscillations [36]. This analysis methodology is suggested to be useful in analysis of the circadian rhythms, such as the flower opening and closing oscillations. As mentioned above, Xu's modified Pressure–Volume curve analysis [31] was used to analyze the osmotic adjustment in response to the sub-irrigation regime. With this analysis, it was elucidated that active accumulation of osmolytes occurred, and consequently leaf turgor potential was maintained and symplast water fraction was increased. The improvements were responsible for the increased photosynthetic activities and promoted plant growth. In addition to the aforementioned improvements, strengthened leaf structure could lead to a higher drought stress resistance, and higher leaf water retention ability was confirmed by the analysis of the excised leaf transpiration declining curve. The lower extent of leaf water loss might be attributed to an increased deposit of wax and lipid compounds into the leaf cuticle and the surface layer. Although the pretty rhythms of flower opening and closing oscillation were found in *E. grandiflorum* plants, the hormonal mechanisms for such rhythms were not among the discussion subjects in the present research, which would be found in the previous report [35] and other documents [41–43]. In conclusion, as one of xerophytophysiological applications, sub-irrigation was a feasible practice that could be used in *E. grandiflorum* culture to improve plant growth as well as flower quality and flowering quality. In future research, we will try more stimulation practices to induce improvements in flower quality and plant growth, elucidate the hormonal mechanisms related with auxin, ethylene, jasmonic acid, and abscisic acid, in addition to salicylic acid, and confirm the gene regulations related with the family of DREB (dehydration resistance element binding) in addition to the *PAL* gene.

## 5. Conclusions

As usual, if a mild stress is imposed to a seedling, the plant can adapt and acclimate to the stress in a later growth period. However, it is difficult to choose a mild stress, such as soil water deficit or salinity, that does not damage the plants. Sub-irrigation is the one method which is just used to maintain soil aeration with a sufficient water supply through the capillary connections. In the present study, sub-irrigation did not damage but rather improved leaf photosynthesis (11% up at $p \leq 0.05$) and consequently promoted biomass production (30% up at $p \leq 0.01$) of *Eustoma grandiflorum* plants. P-V curve analysis showed that sub-irrigation induced osmotic adjustment (32.8 mol m$^3$ at $p \leq 0.01$) and consequently maintained leaf turgor potentials (0.07 MPa up at fully turgid and 0.09 MPa up at midday,

both $p \leq 0.01$). Due to osmotic adjustment, sub-irrigation increased stress tolerance shown by lower osmotic potential (0.08 MPa lower at $p \leq 0.05$) and lower relative leaf water content (0.021 lower at $p \leq 0.05$) at incipient plasmolysis. The water stress resistance was confirmed by analysis of the excised leaf drying curve with a 16% increase ($p \leq 0.01$) in water retention ability (WRA). The main objective of growing *E. grandiflorum* plants was the pretty flowers. As expected, sub-irrigation improved flower quality by increasing anthocyanin concentration (16% up at $p \leq 0.01$). It was also elucidated that an increase in anthocyanin concentration is associated with PAL gene up-regulation (6.7 times up at $p \leq 0.01$), which might be induced by the increase in salicylic acid concentration (13.8 times up at $p \leq 0.01$). Another quality indicator for the flowers is the opening quality, i.e., opening fully during the day and closing completely at night, with a longer opening period. Results in the present study showed that sub-irrigation also improved the flower opening quality in addition to the morphological flower quality. As a unique methodology, the mathematical modeling used to simulate the opening oscillation was feasible. In conclusion, sub-irrigation can be a good stimulation measure to induce xerophytophysiological regulations, aimed at high flower and flowering qualities.

**Author Contributions:** Formal analysis, H.-L.X., J.B. and S.K.; Investigation, T.C.; Methodology, H.-L.X. All authors have read and agreed to the published version of the manuscript.

**Funding:** This study was supported by the Major research project of Jinan University (No. 1420707) and the National Science Foundation for Young Scientists of China (No. 31901490).

**Institutional Review Board Statement:** Not applicable.

**Informed Consent Statement:** Not applicable.

**Data Availability Statement:** Not applicable.

**Conflicts of Interest:** The authors declare no conflict of interest.

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
