# Peer review of "Applications of Xerophytophysiology and Signal Transduction in Plant Production—Flower Qualities in Eustoma grandiflorum Were Improved by Sub-Irrigation"

_sustainability, doi:10.3390/su15021578_

Round 1

Reviewer 1 Report

Dear Authors,

The paper entitled: "Applications of xerophytophysiology and signal transduction in plant production - Flower qualities in Eustoma grandiflorum were improved by sub-irrigation" by Hui-lian Xu, Jianfang Bai, Saneyuki Kawabata, Tingting Chang, is clear and with minor additions can contribute understanding the need to apply sub-irrigation in controlled conditions and contributing to the improvement of growing conditions and the quality of the tested plant species. Further research, it is assumed, can contribute to a wider application and it is necessary to confirm the received results with field trials.

Documented references, 49 in total, are from publications older than 5 years (latest from 2015) which should be reviewed and checked for more recent relevant sources.

The citation of references should be reviewed and harmonized with proper, uniform citations.

Reference marks in the text should be adapted to the instructions for citing them (not superscript).

The reference in line 504 (Patakas and Noitsakis (1997)) needs to be referenced (in square brackets). In the references it is listed as [37].

Reference number 19 does not have the year of publication listed.

As part of the test methods, it was not specified which software was used for statistical data processing.

In all tables in which statistical data are presented, it is necessary to clearly state the meaning of the symbols *, **, ns, and not use, for example, the statement See Table 1 for the symbols and their units (line 524).

When stating the units of measure, it is necessary to separate the numerical value from the unit of measure (example in line 183 (citation 250nm), line 185,  and check in the entire text.

The stated conclusions are descriptive and do not give a clear enough picture of what was obtained from the conducted research and statistical processing of data, but only a statement that could be assumed by setting up a trial, so it should be completed.

Author Response

1)    Documented references, 49 in total, are from publications older than 5 years (latest from 2015) which should be reviewed and checked for more recent relevant sources.
----- Responses.We substituted new articles for most of the olde references. Several reports are our own previous research achievements, and reserved in the references.   
2)    The citation of references should be reviewed and harmonized with proper, uniform citations. 
----- This comment was followed and we harmonized the citations. 
3)    Reference marks in the text should be adapted to the instructions for citing them (not superscript).
      ----- This comment was followed.
4)    The reference in line 504 (Patakas and Noitsakis (1997)) needs to be referenced (in square brackets). In the references it is listed as [37].
----- This comment was followed
5)    Reference number 19 does not have the year of publication listed.
-----This comment was followed.
6)    As part of the test methods, it was not specified which software was used for statistical data processing.
-----This comment was followed and the statistic software was mentioned in the text. 
7)    In all tables in which statistical data are presented, it is necessary to clearly state the meaning of the symbols *, **, ns, and not use, for example, the statement See Table 1 for the symbols and their units (line 524).
-----These comments were followed.
8)    When stating the units of measure, it is necessary to separate the numerical value from the unit of measure (example in line 183 (citation 250nm), line 185,  and check in the entire text.
----- We carefully checked the whole manuscript following the comments.  

The stated conclusions are descriptive and do not give a clear enough picture of what was obtained from the conducted research and statistical processing of data, but only a statement that could be assumed by setting up a trial, so it should be completed.
----- We described the conclusion more clearly and added the quantization of the improvement and the significance levels (p≤ 0.05,  p≤ 0.05 or ns).

Reviewer 2 Report

The manuscript “Applications of xerophytophysiology and signal transduction in plant production - Flower qualities in Eustoma grandiflorum were improved by sub-irrigation” has good and novel results but there are some items that should be considered before acceptance for publication:

Abstract:

-          Some sentences are very long, so, the main subject and meaning was negotiated and missed. It is better that clarify exactly the results.

-          The main purposes as well as hypothesis should be clarify

-          The text should be edited in English language

Introduction:

-      - There are several grammatical problems that should be corrected

-          -The text should be edited completely in English language

-          The main purposes as well as hypotheses should be mentioned

-          The references in the text seems that mentioned as superscripts. They should be written normal in the brackets.

Material and Methods:

-          The text again should be revised and edited in English language

-          The soil sample provided from which upland crop? Specify the crop? Also, give some details regarding the soil characteristics

-          The Subtitle 2.2 is similar to 2.2.1. It is better that 2.2 is different from 2.2.1

-          The subtitles 2.3.1 and 2.3.2 would be better to changed and rewritten

-          Again, there are some typos and grammatical mistakes that should be corrected

-          The references again should be arranged according to journal format

Results:

-          The text should be English edited

-          It is better that show the significant difference among treatments by graphs also

-          The statistical values (ns, * and ** at p<0.05 or p<0.01) are better to indicate under each table

Discussion

-          The discussion is very short and should be improved. Also the results should be interpreted and compared with previous studies and the authors should described well the results.

-          There are several typos and grammatical mistakes that should be corrected.

-          The references at the end of manuscript should be arranged based on journal format

-          The reference 19 mentioned without the year?

-          Reference 27 should be written based on journal format

-          The name of the journals should be written in abbreviation forms based on journal format

Author Response

Abstract:
1)    Some sentences are very long, so, the main subject and meaning was negotiated and missed. It is better that clarify exactly the results.
----- Comments were followed and the quantization of the improvement and the significance levels (p≤ 0.05,  p≤ 0.05 or ns) were added.
-          The main purposes as well as hypothesis should be clarified.
    ----- This comment was followed and the introduction paragraph was improved.
-          The text should be edited in English language.
    ----- We carefully edited the manuscript and corrected grammatical mistakes and typos.  
Introduction:
-      - There are several grammatical problems that should be corrected
 ----- We rechecked carefully the text, corrected mistakes and marked in the text. 
-          -The text should be edited completely in English language
----- We carefully edited the manuscript and corrected grammatical mistakes.    
-          The main purposes as well as hypotheses should be mentioned.
    ----- This comment was followed and the introduction was improved.
-          The references in the text seems that mentioned as superscripts. They should be written normal in the brackets.
    ----- This comment was followed.
Material and Methods:
-          The text again should be revised and edited in English language ---rechecked carefully and corrected. 
-          The soil sample provided from which upland crop? Specify the crop? Also, give some details regarding the soil characteristics --followed and marked in the text (The Andosol soil collected from a field prepared for tomato crop, the soil properties listed). 
-          The Subtitle 2.2 is similar to 2.2.1. It is better that 2.2 is different from 2.2.1 --followed (without the 3ird  serial number)
-          The subtitles 2.3.1 and 2.3.2 would be better to changed and rewritten --followed
-          Again, there are some typos and grammatical mistakes that should be corrected----rechecked carefully and corrected mistakes.
-          The references again should be arranged according to journal format ----corrected 
Results:
-          The text should be English edited ---rechecked carefully and corrected mistakes
-          It is better that show the significant difference among treatments by graphs also. 
------ The data were continuously measured, and each was not an average. Therefore, the is no significant difference indication in the graph. 
-          The statistical values (ns, * and ** at p<0.05 or p<0.01) are better to indicate under each table 
--followed
Discussion
-          The discussion is very short and should be improved. Also the results should be interpreted and compared with previous studies and the authors should described well the results. 
------ followed (compared with the previous studies in tomato, peanut, potato, and fruit trees).
-          There are several typos and grammatical mistakes that should be corrected. 
------ corrected and marked in the text.
-          The references at the end of manuscript should be arranged based on journal format
-          The reference 19 mentioned without the year? ------ corrected.
-          Reference 27 should be written based on journal format  ------ corrected. 
-          The name of the journals should be written in abbreviation forms based on journal format ------ corrected.

Round 2

Reviewer 2 Report

The authors considered all the comments. I think it can be accepted and published.